# Boxhead: A Dataset for Learning Hierarchical Representations

**Yukun Chen**[*]
MPI for Intelligent Systems, Tübingen

**Andrea Dittadi**
Technical University of Denmark

**Frederik Träuble**
MPI for Intelligent Systems, Tübingen

**Stefan Bauer**
KTH Stockholm

**Bernhard Schölkopf**
MPI for Intelligent Systems, Tübingen

## Abstract

Disentanglement is hypothesized to be beneficial towards a number of downstream tasks. However, a common assumption in learning disentangled representations is that the data generative factors are statistically independent. As current methods are almost solely evaluated on toy datasets where this ideal assumption holds, we investigate their performance in hierarchical settings, a relevant feature of real-world data. In this work, we introduce *Boxhead*, a dataset with hierarchically structured ground-truth generative factors. We use this novel dataset to evaluate the performance of state-of-the-art autoencoder-based disentanglement models and observe that hierarchical models generally outperform single-layer VAEs in terms of disentanglement of hierarchically arranged factors.

## 1 Introduction

A common assumption in representation learning is that high-dimensional observations are generated from lower-dimensional generative factors [1]. Disentanglement aims to identify these factors which have statistically independent and semantically meaningful sources [2, 3]. From a causal perspective, disentanglement aims to learn the independent causal mechanisms of the system [4–6]. The possible benefits of disentangling such mechanisms have been investigated in the context of interpretability [7, 8], faster downstream learning [9], fairness [10–12], continual learning [13], and generalization [14–23]. State-of-the-art disentanglement methods are typically based on variational autoencoders (VAEs) with a single latent layer, assume independence of the latent factors [24], and are applied to toy datasets with independent generative factors [3, 25–29]. However, the generative process of real-world data is often hierarchical, i.e. variables describing low-level features are determined by higher-level variables.

VAE-based disentanglement methods with a hierarchical structure in the latent variables such as VLAE [30] and pro-VLAE [31] have been shown to capture to some extent multiple hierarchical levels of detail in data. However, they are not evaluated on datasets where we have a clear understanding of the hierarchical structure underlying the data. This leads to ambiguities when assessing and analyzing the hierarchical structure learned by these models, as there is no ground truth to compare against. Having a dataset with a known, yet simple hierarchical arrangement of generating factors of variation could enable a more controlled and sound quantitative evaluation.

---

[*]Correspondence to: yukunchen113@gmail.com

3rd Workshop on Shared Visual Representations in Human and Machine Intelligence (SVRHM 2021) of the Neural Information Processing Systems (NeurIPS) conference, Virtual.

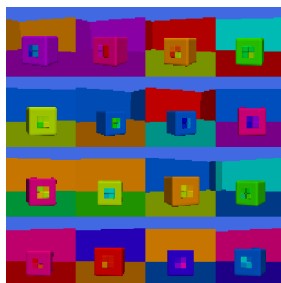 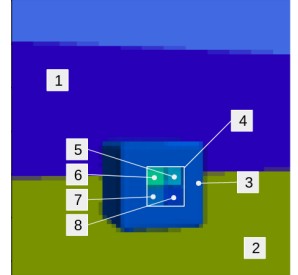 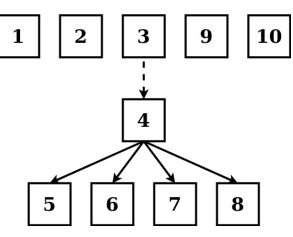

Figure 1: **Left**: Representative example images in our introduced Boxhead dataset. **Center**: Overview of the ground-truth factors: wall hue (1), floor hue (2), cube hue(3), macro-level hue(4), 4x micro-level hue (5,6,7,8), azimuth (9), cube size (10). The macro-level hue is the overall inscribed square color and the micro-level hue factors are the smaller square components of the macro-level hue. **Right**: Hierarchical relationship of factors in the dataset. The dotted arrow only applies to Boxhead 1. The solid arrows applies to all Boxhead variants. Arrow points from macro-level hue factor towards micro-level hue factor(s), where micro-level hue factor values are dependent on the value of the macro-level hue factor.

With this paper we introduce such a hierarchically structured dataset[2] to accurately benchmark representation learning methods in this setting. We use this dataset to evaluate prominent and representative VAE-based disentanglement approaches. In addition, we introduce pro-LVAE, a hierarchical representation learning approach that is competitive with the other methods.

We summarize our contributions as follows:

- We introduce *Boxhead*, a new dataset for learning hierarchically structured representations that has a known hierarchical dependency structure in its factors of variation.

- We propose pro-LVAE, a modification of Ladder VAEs (LVAEs) [32, 33] with a training scheme similar to pro-VLAE.

- We present an empirical study on this new dataset, comparing pro-LVAE with various state-of-the-art disentanglement approaches ($\beta$-VAE, $\beta$-TCVAE, pro-VLAE). We evaluate all models in terms of (i) ELBO, (ii) disentanglement for higher-level (macro) FoVs, and (iii) disentanglement for lower-level (micro) FoVs. Models with a hierarchical structure seem to outperform single-layer VAEs on each of these metrics.

## 2   Background

Variational Autoencoders (VAEs) are latent variable models that aim to model high-dimensional data $\mathbf{x}$ as arising from lower-dimensional latent representation $\mathbf{z}$. VAEs are optimized by defining an approximate posterior distribution $q_\phi(\mathbf{z}|\mathbf{x})$ and maximizing the ELBO:

$$\mathcal{L}(x; \theta, \phi) = \mathbb{E}_{q_\phi(\mathbf{z}|\mathbf{x})}[\log p_\theta(\mathbf{x}|\mathbf{z})] - D_{\mathrm{KL}}(q_\phi(\mathbf{z}|\mathbf{x}) \,\|\, p(\mathbf{z}))$$

which is a lower bound to the log-likelihood $\log p_\theta(x)$, where $p(\mathbf{z})$ is typically a standard isotropic Gaussian. In this work, we will focus on VAEs with (i) one layer of latent variables, (ii) a hierarchy of latent variables given only by the neural architecture, and (iii) hierarchical latent variable models (generative models with a hierarchically structured latent space).

Disentangled representation learning methods aim to learn a representation where the coordinate axes are aligned with the data generative factors, also known as the factors of variation (FoV). Common disentanglement models are single-layer VAEs. They assume statistically independent generative factors and aim for achieving a statistically independent representation by regularizing the latent space. The diagonal covariance prior and stochastic reconstruction loss forces local orthogonality of the coordinate axes, promoting disentanglement of the learned representations [34]. $\beta$-VAEs [27] aim at further encouraging independence in the learned factors by increasing the capacity constraint through $\beta$. Rather than constraining the capacity, models such as $\beta$-TCVAE [3] and FactorVAE [26] explicitly minimize the total correlation of the latent representation, thus maintaining a relatively

---

[2]Code for models and to generate dataset: `https://github.com/yukunchen113/compvae`

better generation quality. It has also been argued that annealing the constraint on the capacity can be beneficial for the reconstruction-disentanglement trade-off in $\beta$-VAEs [2].

Hierarchical disentanglement methods such as VLAE [30], pro-VLAE [31], and SAE [35], model factors that contribute the most to the reconstruction loss in the higher (macro feature) layers, and finer (micro feature) details in the lower layers. However, the latent variables are independent, i.e. the latent prior is fully factorized, so from a probabilistic perspective they are effectively standard VAEs. Notably, pro-VLAE progressively introduces the latent layers during training [31]. This progressive training allows higher latent layers to converge before moving onto the next layers. These higher layers (layer $L$ being the highest) will learn the more abstract macro-level factors, i.e., factors that have a larger effect on the reconstruction loss. This is similar in effect to AnnealedVAE [2, 34]. Finally, hierarchical latent variable models such as LVAE [32], BIVA [33], NVAE [36], SDNs [37], and Very Deep VAEs [38], are more expressive generative models in that their prior does not fully factorize, and they explicitly model statistical dependencies between latents at different layers.

**Defining hierarchy.** Hierarchy is a ubiquitous characteristic in real-world data. However, the exact definition of hierarchy is ambiguous [8, 39–46]. One commonality is that base primitives are used to construct higher-level representations. Roughly, models which focus on learning hierarchical relationships either utilize base primitives to construct higher-level representations [8, 40, 42, 45] or decompose a high-level object into base primitives [39, 41, 44]. The type of hierarchy we will focus on is better defined in causality research as the relationship between micro and macro level factors. Micro-level factors serve as finer detailed factors and macro-level factors serve as the coarse-grained factors. Both macro and micro level factors model the same given system and must be causally consistent [47]. Macro-level factors are the aggregate of a subset of micro-level factors.

## 3  The Boxhead dataset

To foster progress in the learning of hierarchically structured representation, we introduce `Boxhead`, a new dataset with hierarchical factor dependencies in which a cube is placed in a simple 3D scene (see Fig. 1 (left) for a representative sample). This dataset was created with Open3D [48] and PyVista [49] and contains 10 discrete factors of variation: wall hue, floor hue, azimuth, cube size, cube hue, macro-level hue, and four micro-level hue factors. The macro-level hue is the aggregate hue of the square in the center of the cube and the micro-level hues are the hues of the four smaller inscribed squares, their value is dependent on the macro-level hue value. For the specific dataset dependency structure, see Figure 1 (right). See Appendix B.1 for further details on the dataset.

To evaluate hierarchical dependencies, we generate a 64x64 sized image dataset with three separate variants. Each variant has varying levels of dependencies between micro-level and macro-level factors. These variants are named Boxhead 1, Boxhead 2, Boxhead 3 in order of strongest to weakest levels of dependencies. See Appendix B.1 for further details on the dependencies.

As opposed to the 3DShapes dataset [50] which inspired Boxhead, our dataset does not purely have statistically independent factors and instead contains explicit hierarchical structure, an arguably important feature of realistic settings. Compared to CelebA [51], Boxhead contains the ground truth labels of all factors and maintains simplicity for isolated testing of models, a crucial factor given that most disentanglement metrics require completely labeled data.

## 4  Pro-LVAE

We evaluate a set of well-known and commonly used VAE models to show the effects of adding hierarchical architectural components. Single-layer disentanglement VAEs such as $\beta$-VAE or $\beta$-TCVAE do not have hierarchical structure. Pro-VLAE allows for hierarchical sharing of information through the encoder. In addition, we introduce *pro-LVAE*, a novel hierarchical disentanglement model that additionally incorporates statistical dependencies across layers. For pro-LVAE, we apply the same progressive introduction of latent layers as per pro-VLAE on an LVAE. The objective we are

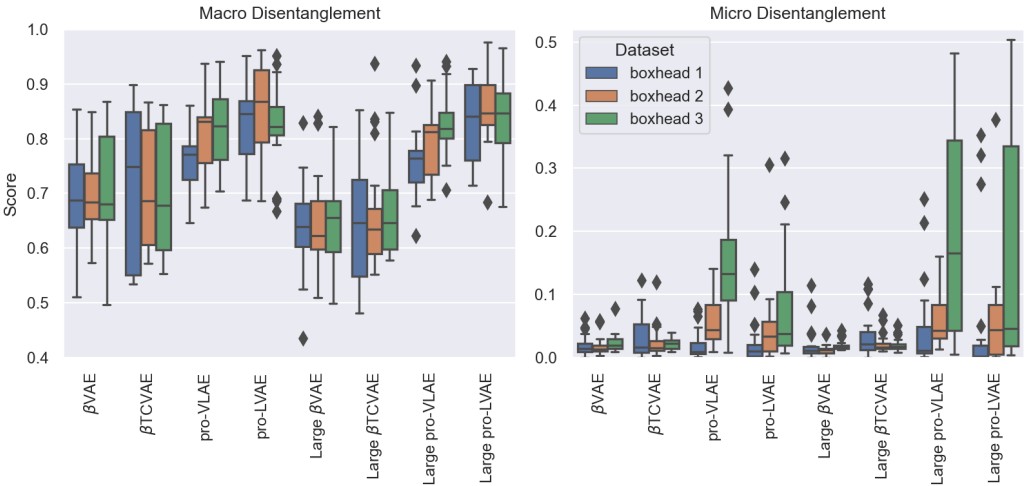

Figure 2: Macro-level and micro-level disentanglement scores for all trained models across all three Boxhead variants. Each box is an aggregation across all $\beta$ and random seed parameters. Hierarchical models appear to perform better than the single-layer models on the Boxhead variants on the micro-level disentanglement score and macro-level disentanglement score. Diamonds represent outliers outside of 1.5 times the interquartile range.

maximizing for pro-LVAE with $L$ latent layers at the $s$-th latent layer introduction step is:

$$\mathcal{L}_{LVAE} = \mathbb{E}_{q(\mathbf{z}|\mathbf{x})}[\log p(\mathbf{x}|\mathbf{z})] - \beta[D_{\mathrm{KL}}(q(\mathbf{z}_L|\mathbf{x}) \,\|\, p(\mathbf{z}_L))$$
$$- \sum_{l=L-s}^{L-1} \mathbb{E}_{q(\mathbf{z}_{>l}|\mathbf{x})}[D_{\mathrm{KL}}(q(\mathbf{z}_l|\mathbf{x}, \mathbf{z}_{>l}) \,\|\, p(\mathbf{z}_l|\mathbf{z}_{>l}))]]$$

where $q(\mathbf{z}_{>l}|\mathbf{x}) = \prod_{i=l+1}^{L} q(\mathbf{z}_i|\mathbf{x}, \mathbf{z}_{>i})$. Here, $\beta$ will constrain the amount of information encoded in the latent layer and will not limit information from the decoder. Similar to pro-VLAE, the various latent layers must contribute different signals to the image reconstruction. This allows for independence between different latent layers, which means we can use the standard disentanglement metrics to evaluate disentanglement across the model. See Fig. 4 in Appendix B for architecture details.

## 5 Experimental Results

**Experimental setup.** To evaluate the benefits of hierarchical inductive biases for disentanglement, we test four different models whose inductive bias allows varying levels of hierarchical information: $\beta$-VAE [27], $\beta$-TCVAE [3], pro-VLAE [31], pro-LVAE. Since model capacity is important for learning micro-level factors, we train every model with small and large capacities. Furthermore, we study four different values of $\beta$ (1, 5, 10, 20) and run all experiments with five random seeds. We train these models on the three variants of the Boxhead dataset. This represents a total of 480 trained models.[3] For fairness, we use 12 latent variables across all models. Ladder models such as pro-VLAE and pro-LVAE have 4 latents for each of their 3 latent layers. We also anneal $\beta$ for the single-layer VAEs and the top layer of the ladder models since the ladder models anneal the capacity by progressively introducing the latent layers. We anneal the top layer of ladder models for 5000 steps. Additionally, since each of the two lower latent layers of the ladder model is progressively introduced over 5000 steps, we anneal the latent layer of the single-layer models over 10000 steps.

**Evaluation metrics.** To measure how well the models disentangle the hierarchical factors, we use the DCI disentanglement metric [52] with 10000 data samples, and compute the score from the resulting importance matrix as detailed in Appendix A. Following [53], we use Scikit-learn's default gradient boosted trees. See Appendix A for a detailed definition. Since the micro- and macro-level data generative factors have a varying impact on the pixel-level reconstruction loss, we split the factors

---

[3]Reproducing these experiments requires approximately 160 GPU days on Quadro RTX 6000 GPUs.

into two separate groups: macro-level and micro-level factors. The macro-level disentanglement score measures disentanglement among the macro-level factors. These factors include floor hue, wall hue, cube hue, azimuth, scale. Similarly, the micro-level disentanglement score measures disentanglement of the micro-level factors.

**Disentanglement.** In Figure 2 we present the macro-level and micro-level disentanglement scores for all trained models across all three Boxhead variants. Hierarchical models perform better than the single-layer models on the Boxhead variants on the micro-level disentanglement score and macro-level disentanglement score. Micro-level disentanglement seems to mirror the underlying dataset structure, where the less correlated dataset variants are easier to disentangle. This is in agreement with the findings in Träuble et al. [12] when

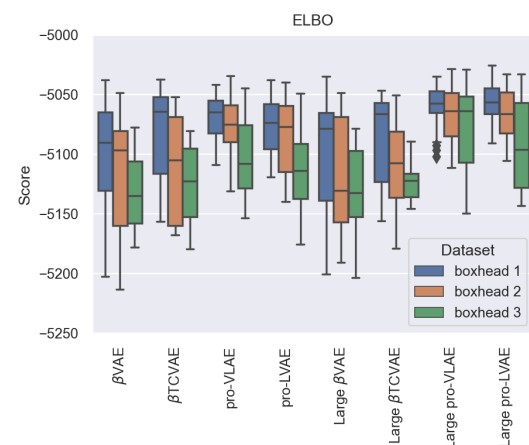

Figure 3: ELBO of all trained models. Each box is an aggregation across all $\beta$ and random seed parameters. Datasets with stronger correlations have higher ELBO: stronger hierarchical dependencies are easier to model.

underlying factors of variation exhibit correlations. Models of larger capacities (low $\beta$, larger architecture) are expected to model finer details. We find that these models exhibit higher micro-level disentanglement scores on this dataset. However, even low-capacity ladder models can disentangle micro-level factors better than high capacity single-layer models. As $\beta$ increases, single-layer model performance on the metrics degrades more than ladder models (see Appendix C).

**ELBO.** Similarly as above, the box plot in Figure 3 shows the respective ELBO scores across all models. Overall we see a clear ordering in ELBO with respect to our three dataset variants. Models trained on strongly correlated variants such as Boxhead 1 consistently have higher ELBO than the models trained on weaker correlated variants such as Boxhead 3. Even though disentanglement is worse, stronger hierarchical dependencies are easier to model. VAEs with hierarchical dependencies have higher ELBO scores compared to their single-layer counterparts. As $\beta$ increases, the ELBO score decreases for all models. However, this happens at different values of $\beta$ for different models (See Appendix C). For higher-capacity VLAEs and LVAEs, the ELBO starts degrading at a $\beta$ of around 10, low capacity VLAEs and LVAEs at a $\beta$ of around 5, and the ELBO degrades at any $\beta$ for single layer VAEs. This shows that the ELBO for VAEs with hierarchical dependencies and higher capacities are more robust to higher values of $\beta$, which can provide better disentanglement. The ELBO is lower for single-layer VAEs because the capacity constraint is applied across a given layer in the model. When enforcing higher capacity constraints on single layer VAEs, micro-level factors collapse when competing with macro-level factors. On the other hand, VAEs with multiple latent layers allow factors to compete at their respective levels. However, with high $\beta$ (such as $\beta = 20$), posterior collapse starts to happen even with multiple latent layers.

## 6  Related Work

There have been several methods proposed towards achieving and defining disentanglement. InfoGAN [25] forms disentangled representations in GANs by maximizing the mutual information between the noise variables and the observations with an auxiliary network. Locatello et al. [53] evaluated state-of-the-art single-layer VAE disentanglement approaches, showing that unsupervised disentanglement is impossible without inductive biases, and highly sensitive to hyperparameters and random seeds. Shu et al. [54] use consistency and restrictiveness to define disentanglement, qualities which also directly represent predictability, a quality seen in humans [55]. Recent work has shown that current disentanglement models particularly struggle on data with correlated factors [12] in the unsupervised setting but can be mitigated with weak labels [19]. There also exists recent work towards hierarchical disentanglement by Ross and Doshi-Velez [46]. Their definition of hierarchy differs from ours as they do not have an equivalent macro model of the complete system, which we represent through aggregation. A macro factor in our definition is the aggregate representation of a subset of micro factors.

# 7 Conclusion

We introduced a new dataset to analyze the disentanglement of hierarchical factors and evaluated state-of-the-art disentanglement models to show the benefits of hierarchical inductive biases. We show that hierarchical models outperform single-layer VAEs in disentangling micro- and macro-level factors, and suggest a better ELBO. The performance of hierarchical models over single-layer models becomes more prominent as the capacity constraint ($\beta$) increases. Future work includes identifying inductive biases for hierarchical latent space models to better disentangle hierarchical factors, especially micro-level factors which contribute less to the reconstruction loss. An additional promising direction is the application of these hierarchical models for realistic environments within dynamic and interactive observation settings: learning varying levels of detail could be helpful towards disentanglement in scenarios where the observer moves closer to an object of interest. Another interesting area of future research is on metrics for evaluating hierarchical relationships.

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

# A Model Objectives and Metric Definition

$\beta$**VAE Objective:** The objective we are maximizing for $\beta$-VAE is defined below, with $\beta > 1$:

$$\mathcal{L}(x; \theta) = \mathbb{E}_{q(\mathbf{z}|\mathbf{x})}[\log p(\mathbf{x}|\mathbf{z})] - \beta D_{\mathrm{KL}}(q(\mathbf{z}|\mathbf{x}) \, \| \, p(\mathbf{z}))$$

$\beta$**TCVAE Objective:** The $\beta$-TCVAE objective we are maximizing is:

$$\mathcal{L}_{TCVAE} = \mathbb{E}_{q(\mathbf{z}|\mathbf{x})}[\log p(\mathbf{x}|\mathbf{z})] - I_q(z; x) - \beta D_{\mathrm{KL}}(q(\mathbf{z}) \, \| \, \prod_j q(\mathbf{z}_j)) - \sum_j D_{\mathrm{KL}}(q(\mathbf{z}_j) \, \| \, p(\mathbf{z}_j))$$

**Pro-VLAE Objective:** At the $s$th latent layer introduction step, the objective we are maximizing for pro-VLAE is:

$$\mathcal{L}_{VLAE} = \mathbb{E}_{q(\mathbf{z}_L, \mathbf{z}_{L-1}, ..., \mathbf{z}_{L-s}|\mathbf{x})}[\log p(x|\mathbf{z}_L, \mathbf{z}_{L-1}, ..., \mathbf{z}_{L-s})] - \beta \sum_{l=L-s}^{L} D_{\mathrm{KL}}(q(\mathbf{z}_l|\mathbf{x}) \, \| \, p(\mathbf{z}_l))$$

Where $L$ is the number of latent layers.

**DCI Disentanglement Score:** The DCI disentanglement score measures if a latent element contains information about a given data generative factor. The disentanglement score is calculated by the equation: $\sum_i \rho_i(1 - H(P_i))$. Given the $i$th latent representation and the $j$th data generative factor, $P_{ij} = R_{ij} / \sum_{k=0}^{K-1} R_{ik}$. $R$ is the importance matrix. $H(P_i)$ calculates the entropy of $P_i$. $\rho_i$ is a multiplier used to account for irrelevant and dead units in the latent representation.

# B Further Model and Dataset Details

## B.1 Boxhead Dataset Specifications

For all Boxhead datasets, the hue is bounded by [0, 1], using the HLV color scheme. The distribution that generates macro-level hue is discretized into 7 values. The distributions for the micro-level hues is discretized into 4 values. Other factors of variation are discretized into 10 values. Discretization happens on the distribution before the addition of information from other factors.

**All Boxhead variants.** All boxhead variants have the following specifications:

- wall hue $\sim U(0, 1)$
- floor hue $\sim U(0, 1)$
- cube hue $\sim U(0, 1)$
- scale $\sim U(1,\ 1.25)$
- azimuth $\sim U\left(-\frac{\pi}{6}, \frac{\pi}{6}\right)$

**Boxhead 1.** The Boxhead 1 variant has the additional following specifications:

- macro-level hue $\sim \mathcal{N}(\text{cube hue}, 0.2)$
- $i$th micro-level hue $\sim U(\text{macro-level hue} - 0.1, \text{macro-level hue} + 0.1)$ for $i = 1, 2, 3, 4$

**Boxhead 2.** The Boxhead 2 variant has the additional following specifications:

- macro-level hue $\sim U(0, 1)$
- $i$th micro-level hue $\sim \mathcal{N}(\text{macro-level hue}, 0.1)$ for $i = 1, 2, 3, 4$

**Boxhead 3.** The Boxhead 3 variant has the additional following specifications:

- macro-level hue $\sim U(0, 1)$
- $i$th micro-level hue $\sim \mathcal{N}(\text{macro-level hue}, 0.2)$ for $i = 1, 2, 3, 4$

## B.2  Model Architecture Details

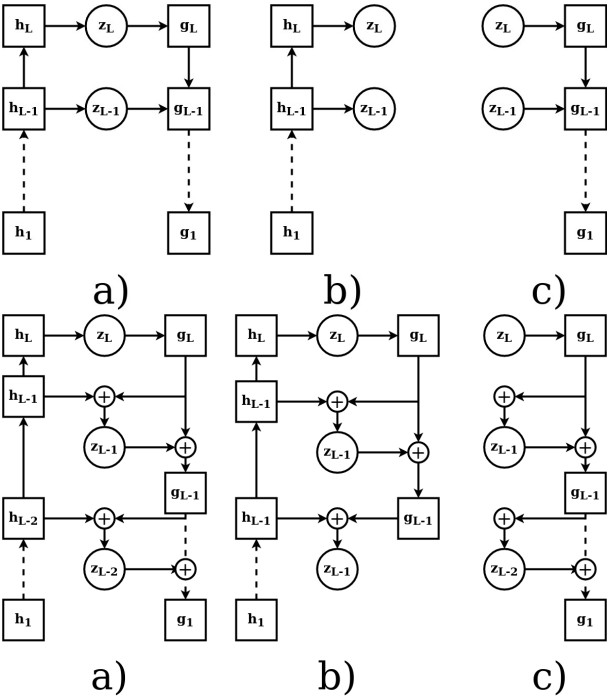

Figure 4: Two rows of architectures. Top row: pro-VLAE, Bottom Row: pro-LVAE. $h_l$ are convolutional blocks, $\mathbf{z}_l$ is the latent representations. $\bigoplus$ denotes concatenation. a) shows the process during reconstruction. b) shows the encoding step. c) shows the decoding step.

**Network Architecture Specifications** Here, we detail the network architecture for the models. All models have equivalent network layers. Ladder models have additional ladder layer specifications. Each ladder layer contains an encoding section and a decoding section. There are two architectures; a small architecture and a large architecture. Default activation is leaky ReLU, decoder output activation is sigmoid, encoder activation is linear (no change).

| | | |
|---|---|---|
| $h_1$ | conv | [64,4,2] |
| | bn | |
| $h_2$ | conv | [128,4,2] |
| | bn | |
| $h_3$ | conv | [256,2,2] |
| | bn | |
| | conv | [512,2,2] |
| | bn | |
| | flatten | |
| | dense | [1024] |
| | bn | |
| $g_3$ | dense | [1024] |
| | bn | |
| | dense | [4*4*512] |
| | bn | |
| | reshape | [4,4,512] |
| | conv | [256,4,2] |
| | bn | |
| $g_2$ | conv | [256,4,1] |
| | bn | |
| | conv | [128,4,2] |
| | bn | |
| $g_1$ | conv | [128,4,2] |
| | bn | |
| | conv | [64,4,1] |
| | bn | |
| $g_0$ | conv | [3,4,2] |

Table 1: Small architecture ("bn" denotes batch normalization).

| | | |
|---|---|---|
| $h_1$ | conv | [64,1,2] |
| | resnet | [64,4,1], bn, [64,4,1], bn |
| $h_2$ | conv | [128,1,2] |
| | resnet | [128,4,1], bn, [128,4,1], bn |
| $h_3$ | conv | [256,1,2] |
| | resnet | [256,2,1], bn, [256,2,1], bn |
| | conv | [512,2,2] |
| | bn | |
| | flatten | |
| | dense | [1024] |
| | bn | |
| $g_3$ | dense | [1024] |
| | bn | |
| | dense | [4*4*512] |
| | bn | |
| | reshape | [4,4,512] |
| | conv | [256,4,2] |
| | bn | |
| | conv | [256,1,2] |
| | resnet | [256,4,1], bn, [256,4,1], bn |
| | conv | [128,4,1] |
| $g_2$ | conv | [128,1,1] |
| | resnet | [128,4,1], bn, [128,4,1], bn |
| | conv | [64,4,2] |
| $g_1$ | conv | [64,1,1] |
| | resnet | [64,4,1], bn, [64,4,1], bn |
| $g_0$ | conv | [3,4,2] |

Table 2: Large architecture ("bn" denotes batch normalization).

| | | |
|---|---|---|
| encoding | conv | [128,4,2] |
| | bn | |
| | conv | [256,4,1] |
| | bn | |
| | flatten | |
| | dense | [1024] |
| | bn | |
| decoding | dense | [1024] |
| | bn | |
| | dense | [prod(output shape)] |
| | bn | |
| | reshape | [output shape] |

Table 3: Small architecture, latent layer 1 and 2 (lowest and middle layers).

| | | |
|---|---|---|
| encoding | conv | [128,4,2] |
| | bn | |
| | conv | [256,4,1] |
| | bn | |
| | flatten | |
| | dense | [1024] |
| | bn | |
| decoding | dense | [1024] |
| | bn | |
| | dense | [prod(output shape)] |
| | bn | |
| | reshape | [output shape] |

Table 4: Large architecture, latent layers 1 and 2 (lowest and middle layers).

# C  Additional Experimental Results

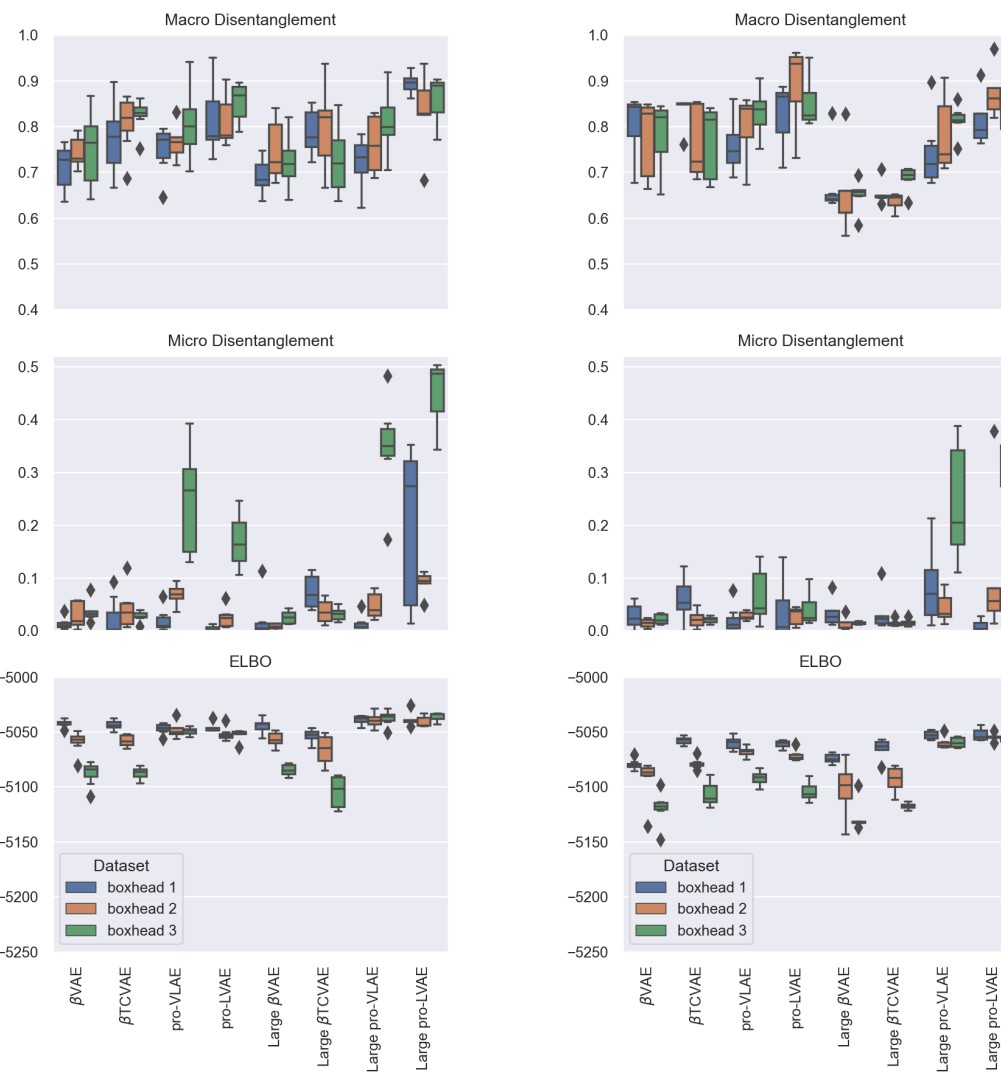

Figure 5: Box plots of evaluation metrics for all models with $\beta = 1$.

Figure 6: Box plots of evaluation metrics for all models with $\beta = 5$.

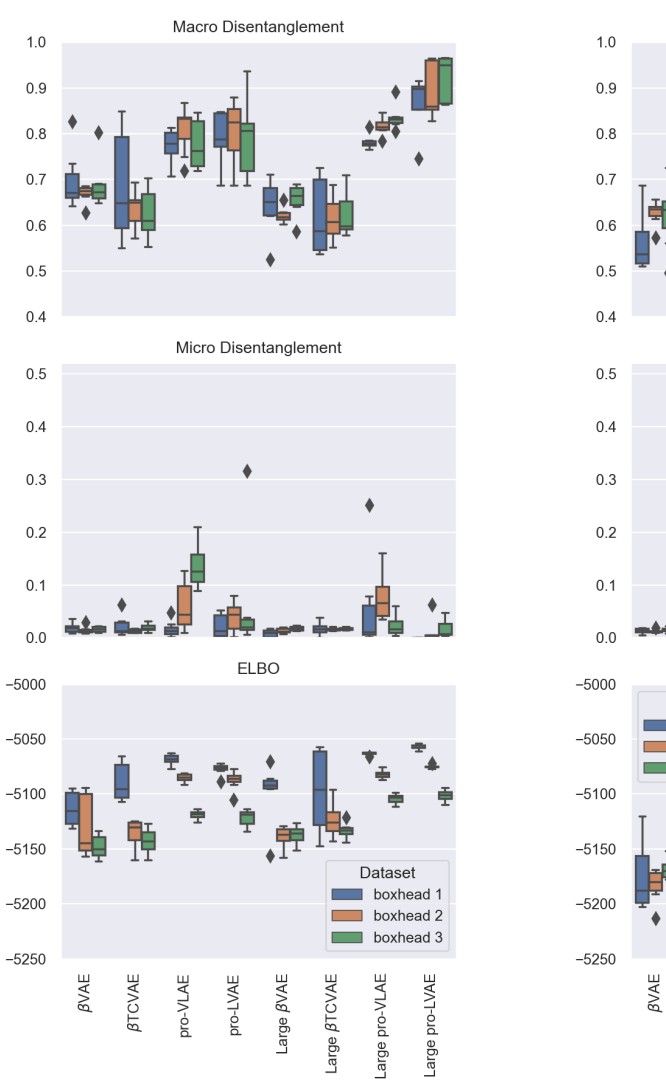

Figure 7: Box plots of evaluation metrics for all models with $\beta = 10$.

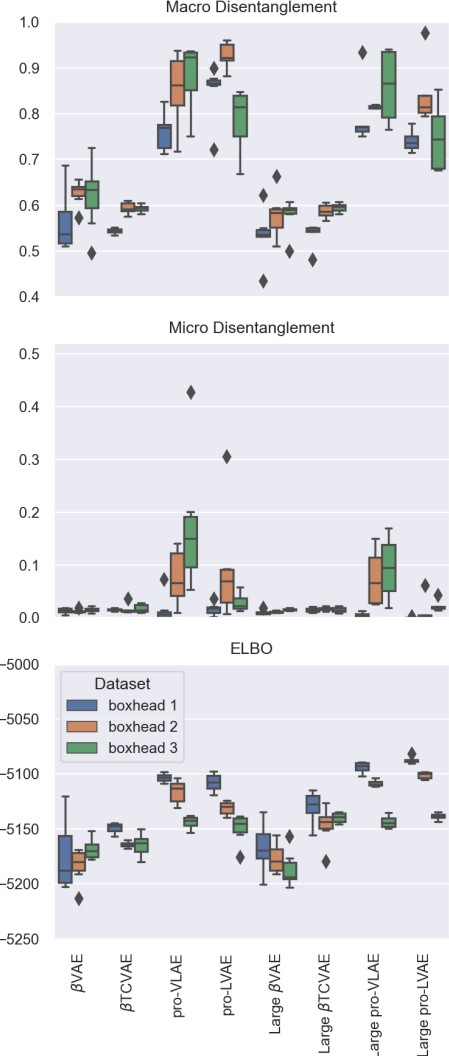

Figure 8: Box plots of evaluation metrics for all models with $\beta = 20$.

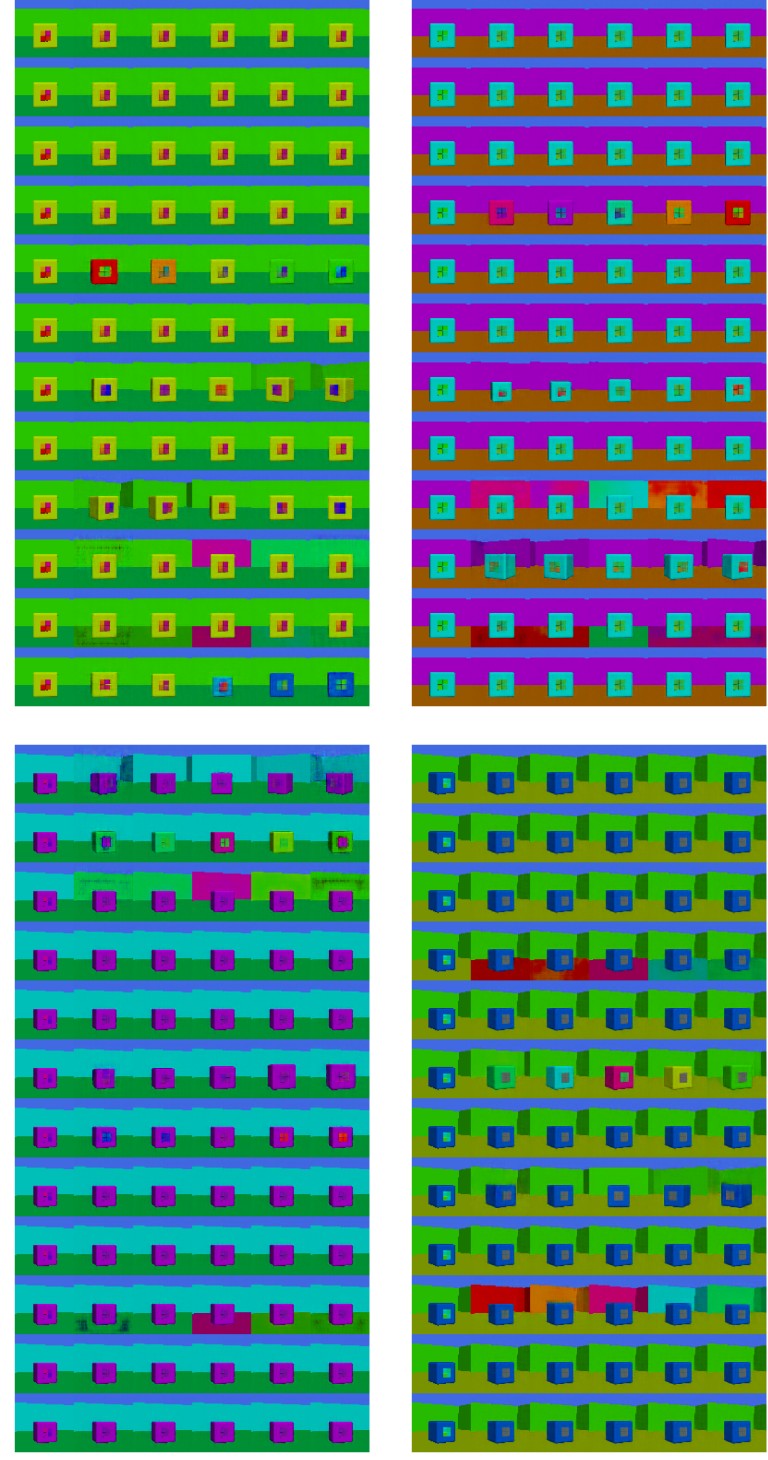

Figure 9: Latent traversal of top Micro Disentanglement score model among $\beta$VAEs (top) and $\beta$TCVAEs (bottom) with a small (left) and large architecture (right). $\beta = 1$, Boxhead 3. Rows are the latents. Leftmost column is the ground truth image. The next 5 columns is a traversal from -3$\sigma$ to 3$\sigma$, where $\sigma$ is the standard deviation of the prior for the layer being traversed.

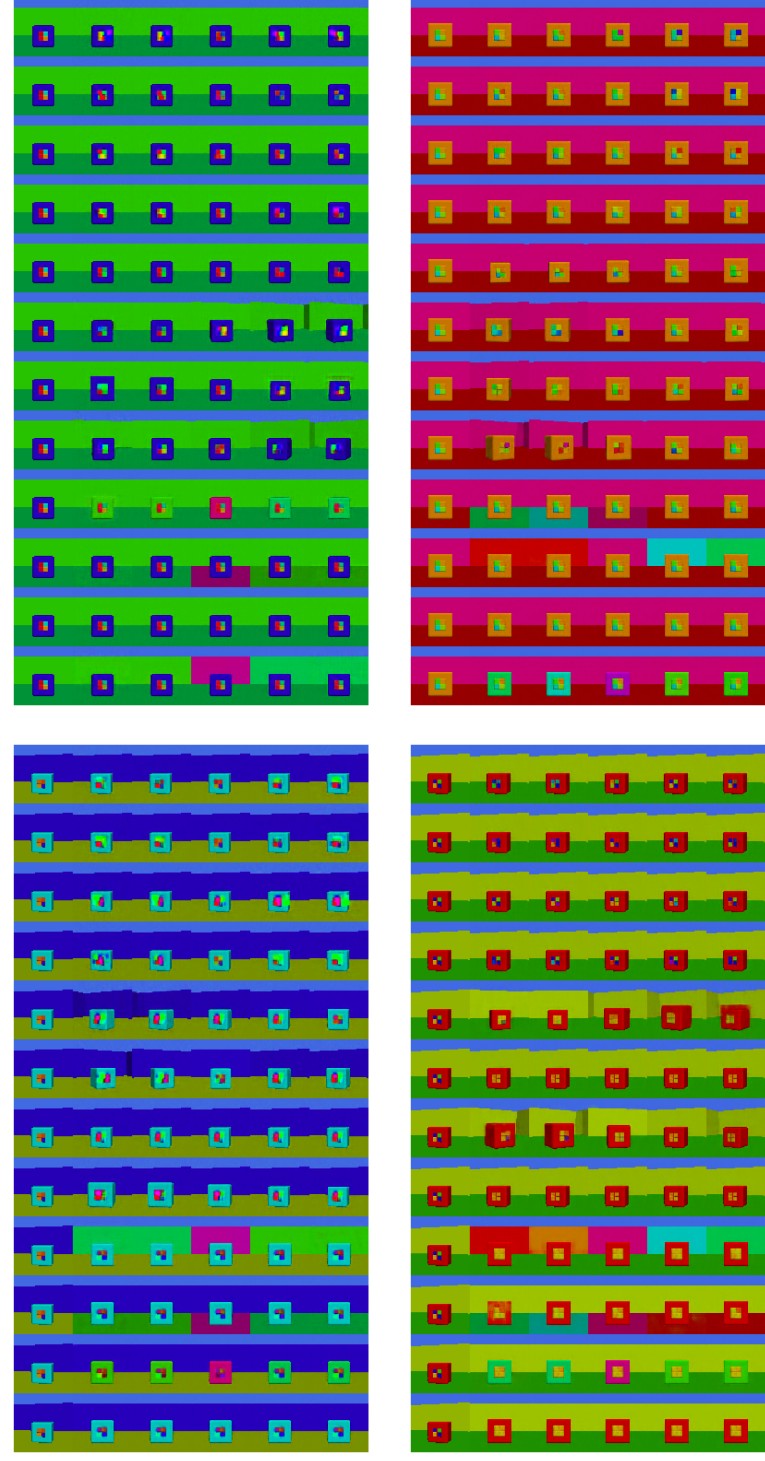

Figure 10: Latent traversal of top Micro Disentanglement score model among pro-VLAEs (top) and LVAEs (bottom) with a small (left) and large architecture (right) pro-VLAEs with a small architecture. $\beta = 1$, Boxhead 3. Rows are the latents, each group of four rows is a layer, with the highest layer (L) at the bottom. Leftmost column is the ground truth image. The next 5 columns is a traversal from -3$\sigma$ to 3$\sigma$, where $\sigma$ is the standard deviation of the prior for the layer being traversed.

