# OpenReview forum: "Boxhead: A Dataset for Learning Hierarchical Representations"
_NeurIPS.cc/2021/Workshop/SVRHM — SVRHM 2021 Poster_

### Official Review · Reviewer_qX97 · 2021-10-29

**Rating:** 8
**Confidence:** 3

**Review:**

This paper introduces a dataset 'Boxhead' for studying disentanglement when the latent variables have explicit dependencies. This dataset permits quantitative evaluation of disentanglement in learned representations in scenarios that are more realistic of real-world datasets (i.e. it doesn't assume independent generating factors). The authors also performed extensive experiments, and quantitatively evaluated the disentanglement at two different hierarchical levels. The authors also fused a modern architecture and optimization schedule leading to what they termed 'pro-VLAE'.

Overall this is a simple but good paper that, to the best of my knowledge, fills an important need in the literature in creating a quantitative dataset to evaluate disentanglement when the latent factors are dependent. Moreover, it represents a step to creating more challenging and realistic datasets for evaluating disentanglement when ground-truth factors are known. The paper was also quite well written, and the experiments performed were thorough and convincing.

I have the following points:
- In the definition of the dataset, it was not clear whether the distribution that the latent variables were sampled from were continuous or discrete. Certain disentanglement metrics work better on discrete data. I suggest that the authors clarify this decision in the paper.

- The authors mention in the discussion that the creation of metrics to evaluate hierarchical disentangled representations is a direction of future research, but it was not mentioned why other metrics are not ideal for measuring disentanglement in these scenarios. Are there any other disentanglement metrics other than DCI that work well for evaluating these hierarchical disentangled representations?

Minor
- Also, the authors should clarify the notation for "micro - level hue" in the Appendix. This read as "macro minus level hue".

---

### Official Review · Reviewer_Ath1 · 2021-10-31
**A good hierarchical dataset for evaluating hierarchical disentanglement methods**

**Rating:** 7
**Confidence:** 3

**Review:**

Summary: In this paper, the authors introduce a family of datasets with various degrees of hierarchical structure, and show that hierarchical disentanglement methods (pro-VLAE, pro-LVAE) perform better on these datasets than vanilla methods (b-VAE, TCVAE).

Recommendation: I recommend acceptance of this paper, as to my knowledge it fills an important gap in the literature of hierarchical disentanglement. Indeed, hierarchical disentanglement methods have not been tested on artificial hierarchical datasets for which we know the ground-truth factors of variations.

Suggestions to improve the paper:
- The existing (VLAE) and proposed (LVAE) hierarchical models are not described in sufficient detail to understand how they work. How are the different latent layers connected exactly? What ensures that they are able to extract hierarchical factors of variations? Do these models come with theoretical guarantees?
- The proposed new model (LVAE) does not seem to work much better than the other hierarchical models. Could the authors comment on this? The motivation for introducing this new model is also unclear to me. What is the advantage of this "novel hierarchical disentanglement model that additionally incorporates statistical dependencies across layers"?
- fig2 : what are the diamonds?
- An alternative hierarchical dataset is proposed in Saxe et al. 2018: https://www.pnas.org/content/116/23/11537.short. It would be interesting to find a description of the potential conceptual differences and the advantages of the proposed dataset over this other dataset to evaluate hierarchical disentanglement methods.

---

### Decision · Program_Chairs · 2021-11-02

Accept (Poster)